# Current Strategies to Guide the Antiplatelet Therapy in Acute Coronary Syndromes

**DOI:** 10.3390/ijms25073981

**Published:** 2024-04-03

**Authors:** Isabella Russo, Carola Griffith Brookles, Cristina Barale, Elena Melchionda, Amir Hassan Mousavi, Carloalberto Biolè, Alessandra Chinaglia, Matteo Bianco

**Affiliations:** 1Department of Clinical and Biological Sciences, University of Turin, I-10043 Turin, Italy; isabella.russo@unito.it (I.R.); cristina.barale@unito.it (C.B.); elena.melchionda@unito.it (E.M.); 2Cardiology Division, San Luigi Gonzaga University Hospital, I-10043 Orbassano, Italy; carola.griffithbrookles@unito.it (C.G.B.); amir.mousavi@edu.unito.it (A.H.M.); carloalberto.biole@gmail.com (C.B.); a.chinaglia@sanluigi.piemonte.it (A.C.); 3Department of Medical Sciences, University of Turin, I-10124 Turin, Italy

**Keywords:** antiplatelet therapy, acute coronary syndromes, platelet function testing, genetic testing

## Abstract

The role of antiplatelet therapy in patients with acute coronary syndromes is a moving target with considerable novelty in the last few years. The pathophysiological basis of the treatment depends on platelet biology and physiology, and the interplay between these aspects and clinical practice must guide the physician in determining the best therapeutic options for patients with acute coronary syndromes. In the present narrative review, we discuss the latest novelties in the antiplatelet therapy of patients with acute coronary syndromes. We start with a description of platelet biology and the role of the main platelet signal pathways involved in platelet aggregation during an acute coronary syndrome. Then, we present the latest evidence on the evaluation of platelet function, focusing on the strengths and weaknesses of each platelet’s function test. We continue our review by describing the role of aspirin and P2Y12 inhibitors in the treatment of acute coronary syndromes, critically appraising the available evidence from clinical trials, and providing current international guidelines and recommendations. Finally, we describe alternative therapeutic regimens to standard dual antiplatelet therapy, in particular for patients at high bleeding risk. The aim of our review is to give a comprehensive representation of current data on antiplatelet therapy in patients with acute coronary syndromes that could be useful both for clinicians and basic science researchers to be up-to-date on this complex topic.

## 1. Introduction

Dual antiplatelet therapy (DAPT) refers to the combination of two different agents whose effect is the inhibition of platelets’ activation and aggregation, which ultimately results in thrombus formation. Nowadays, the term identifies the simultaneous use of aspirin and a P2Y12 inhibitor (P2Y12i), a cornerstone in the treatment of acute coronary syndromes (ACS) [1,2].

DAPT regimens have enormously evolved in the latest years in the field of ACS and percutaneous coronary interventions (PCI), as advances in stent technology, together with progressive ageing and rising frailty in patients, have shifted attention from thrombotic to hemorrhagic risk.

The aim of the review is to outline the latest evidence regarding DAPT, which has emerged from both observational studies and randomized clinical trials and has been expressed in the most recent guidelines, providing insight into the mechanisms of action of the most common antiplatelet agents as a means to understand the advantages and pitfalls of different pharmacological options according to the patient’s ischemic and hemorrhagic risk. Moreover, after a brief description of platelet function and response, specifically to adenosine diphosphate (ADP) and the different laboratory approaches monitoring the anti-aggregating therapy, it will focus on the role of laboratory platelets’ screening tests in guiding the choice of the best antithrombotic regimen.

## 2. Platelet Biology and Function

Platelets are the smallest cells in circulating blood, averaging only 2.0 to 5.0 μm in diameter, 0.5 μm in thickness, and 6 to 10 femtoliters in volume. Our body produces about 100 billion platelets every day, but only a small fraction of these are consumed in hemostatic processes, while the majority are cleared by the spleen and the reticuloendothelial system. Their life span in circulation is only 10 days [3,4,5]. The platelet plasma membrane is relatively smooth compared to that of leukocytes in circulating blood, a lipid bilayer not different from the membrane covering other cells and with an extremely important role in the acceleration of clotting, a function not shared by the other blood circulating cells [6]. Platelets are primarily and appropriately considered to function to sustain the hemostatic mechanisms, both by preserving the integrity of blood vessel walls and by contributing to the blood coagulation process.

An intact endothelium prevents thrombus formation mainly by the release of nitric oxide (NO) and prostacyclin (PGI2), the two major physiological antiplatelet agents [7,8,9]. The antiplatelet properties of NO and PGI2 depend on their ability to activate the cyclic guanosine monophosphate (cGMP)/protein kinase G (PKG) [10] and the cyclic adenosine monophosphate (cAMP)/protein kinase A (PKA) [11,12] pathways, respectively.

However, if the activation of these pathways is impaired, platelet sensitivity to agonists increases, contributing to platelet activation [13,14,15,16,17,18,19]. Other specific receptors normally expressed by endothelial cells contribute to controlling coagulation by binding thrombin and switching its coagulant into anticoagulant effects. For instance, thrombomodulin in conjunction with the protein C receptor leads to the thrombin-catalyzed activation of protein C to form activated protein C (APC), a circulating serine protease that irreversibly inactivates the coagulant factors Va and VIIIa [20,21]. Furthermore, the endothelial constitutive expression of the inhibitor of tissue factor (TF) pathway, which binds and inhibits the factor VIIa/TF complex, prevents the initiation of the extrinsic coagulation pathway [22].

Platelet involvement in repairing damaged walls is accomplished by the expression of surface receptors recognizing exposed components normally expressed by connective tissue but covered by endothelial cells and/or by the release of soluble factors released by endothelial and other connective tissue cells [23]. When damage is detected, the platelet response is rapid. They attach, change shape, and spread over the damaged area. The activation of platelets is contextual to: (i) secretion, which moves adhesive receptors onto the platelet surface; (ii) release of agonist compounds, which attract and activate other platelets; (iii) the activation of biochemical synthetic pathways leading to activation of cyclooxygenase-1 (COX-1) following production and release of thromboxane (TX), a potent platelet agonist; and (iv) the activation of the platelet surface integrins, including glycoprotein (GP) IIb-IIIa, the major platelet integrin binding fibrinogen, and von Willebrand factor (vWF). GPIIb-IIIa receptor is in an inactive form in the resting platelets, whereas after platelet activation, signals generated inside platelets modify GPIIb-IIIa conformation, permitting ligand binding [24].

The first event recognized when platelets make contact with a damaged surface is the change from the discoid to a rounded or spheroid shape. Next, filopods (fingerlike projections) are elaborated from the cell periphery, then platelets flatten over surfaces, granules and organelles are squeezed into the center of the platelet, and many lamellae are extended. The coalescence of lamellae forms a single, large circumferential lamella, thus greatly increasing the surface area of the cell. For platelet rounding, a transient rise of cytosolic calcium, following receptor ligation, plays a crucial role. Calcium mobilization is mediated by the activation of phospholipase C, an enzyme that hydrolyzes the membrane-bound phosphatidylinositol 4,5-bisphosphate [PI(4,5)P_2_] to form the second messengers, diacylglycerol (DAG) and inositoltriphosphate (IP3) [25,26]. Phospholipase C (PLC) is activated by the βγ-subunit of trimeric G-proteins, which couples it to the serpentine receptors on the platelet surface, including the protease-activated receptor (PAR) family (PAR-1, PAR-4), ADP receptors (P2Y1 and P2Y12), and the serotonin receptor. Calcium can also be mobilized through PLC activation coupled to the immunoreceptor tyrosine-based activation motif (ITAM) domains of the γ-subunit associated with the collagen receptor, GPVI, and the FcγRIIa receptor [27].

GP receptors are necessary to facilitate platelet adhesion to a damaged surface to promote platelet aggregation, activation, and the process of clot retraction [23]. The principal glycoprotein receptors are the GPIb-IX-V and the GPIIb-IIIa complex, two mobile receptors extremely important for their function in hemostasis. Resting platelets display about 25,000 GPIb-IX and about 80,000 GPIIb-IIIa receptors distributed on the outside surface and lining channels of the open canalicular system. The exposure of the vascular subendothelium determines the immediate attachment of GPIb-IX to VWF, which covers collagen fibers. Two collagen receptors, GPVI and integrin α2β1, stabilize the attachment, and GPVI together with GPIb-IX activates GPIIb-IIIa, a complex that binds fibrinogen and fibronectin in the damaged site (Figure 1).

GPIb-IX binding to vWF also triggers actin filament formation and cytoskeletal assembly in newly adherent platelets, together with the release of granule content. Additional platelets adhere to spread cells on the damaged surface, binding fibrinogen via newly expressed GPIIb-IIIa complexes, thus leading to platelet aggregates.

Platelets contain three major types of granules, including α granules, dense bodies (δ granules), and lysosomes [28]. The number of α granules in platelets mainly depends on the size of the cell. There are usually 40 to 80 α granules per platelet, but large and giant cells may have well over 100 α granules. Proteins contained in α granules are synthesized by the megakaryocytes and represent platelet-selective proteins, including coagulation factor V, thrombospondin, P-selectin, and vWF. Other proteins, such as fibrinogen, are synthesized in other cells and taken up by platelets. Platelet δ granules (4 to 8 per platelet) are smaller than the α granules and have high morphological variability. δ granules are rich in adenine nucleotides, including adenosine triphosphate (ATP) and ADP, serotonin, pyrophosphate, calcium, and magnesium. Human platelets contain a few lysosomes (no more than 3 lysosomes per platelet), containing at least 13 acid hydrolases, cathepsin D and E, lysosome-associated membrane protein 2 (LAMP2), and CD63 [29].

Proteomic-based studies revealed thousands of proteins in platelets from healthy subjects, and the number is expanding due to the improvement in efficiency and sensitivity of these types of investigations [30]. Although platelet proteomes are endogenously synthesized by megakaryocytes at various stages of the differentiation process, emerging evidence shows that a variety of factors can induce platelets to adapt their proteome after they bud from megakaryocytes with adjustments of repertoire and expression that include constitutive and signal-dependent protein synthesis, posttranslational modifications, and protein degradation [31]. Variations in the platelet proteome have been found in patients at increased risk for thrombosis or bleeding [32,33]. Proteomic changes have also been observed in platelets from patients with ACS [34]. Indeed, more than 20 proteins mainly involved in cytoskeletal and signaling events were differentially regulated in platelets from non-ST-elevation acute coronary syndrome (NSTE-ACS) patients. Furthermore, in comparison to patients with stable coronary disease, those with acute coronary ischemia showed impaired expression of proteins associated not only with the cytoskeleton but also glycolysis and the antioxidant system. Drugs can also interfere with the platelet proteome in patients with coronary artery disease. Differential expressions of proteins involved in energetic metabolism and oxidative stress were found in patients stratified as resistant and sensitive to aspirin treatment. Similarly, the proteome in platelets from patients in treatment with antihypertension drugs shows some modifications [35].

## 3. Role of P2Y12 and P2Y1 in ADP-Induced Platelet Activation

ADP is a week agonist with a crucial role in amplifying platelet responses to other agonists by acting in an autocrine manner via its secretion from platelet δ granules. The main ADP receptors on platelets involve two separate components, P2Y12 and P2Y1, belonging to the seven transmembrane families, with P2Y12 much more expressed than P2Y1 [36] (Figure 2).

The P2Y12 receptor is a G-protein-coupled receptor (GPCR) coupled primarily to G_i_ proteins [37], and its activation by ADP promotes the agonist-induced release of δ granule, pro-coagulant activity, and thrombus formation [38]. The inhibition of adenylate cyclase (AC), following the decrease in cAMP synthesis, and the phosphatidylinositol 3-kinase (PI-3K) pathway activation [39] with the consequent increase in protein kinase B (AKT) phosphorylation are the main mechanisms by which ADP binding to P2Y12 eventually leads to platelet activation and aggregation [40]. The Gq-coupled P2Y12 receptor activation mediates a transient cytoplasmic calcium increase, shape change, and reversible aggregation. Indeed, a normal aggregation response to ADP requires the concomitant activation of the Gq and Gi pathways mediated by P2Y1 and P2Y12 [40,41], one responsible for platelet shape change and aggregation and the other for inhibition of adenylyl cyclase, respectively. Although some studies have demonstrated the role of TXA2 generation in ADP-induced platelet aggregation [42], others have shown that in patients with P2Y12 deficiency, TXA2 synthesis is preserved [43,44]. The disparity in these results might be due to the numerous challenges in investigating the signaling pathways involved.

Although ADP per se is unable to induce the release of delta granule content [40], it significantly contributes to amplifying platelet secretion induced by other agonists, including TXA2 and thrombin [45]. The pivotal role of the P2Y12 receptor in mediating platelet response to other agonists comes from studies in human platelets congenitally lacking the P2Y12 receptor and platelets from P2Y12 receptor knockout mice, which show impaired responses to the TXA2 mimetic U46619, epinephrine, and low concentrations of collagen and thrombin [46,47,48]. These findings support the role of the interplay between the platelet ADP receptors in the pathogenesis of platelet thrombi, as suggested by the finding that their concomitant inhibition completely abolished shear-induced platelet aggregation [49]. Since the reduction in platelet aggregation is crucial to preventing cardiovascular events, including myocardial infarction, a number of compounds or active metabolites that antagonize ADP-induced platelet aggregation by blocking the P2Y12 receptor have been conceived [50].

## 4. Different Approaches for Testing Platelet Function

Since aggregation of platelets is a critical event to maintain hemostasis within the vasculature and its dysregulation can cause bleeding or thrombosis, several platelet function tests can be performed.

### 4.1. Light Transmission Aggregometry

To better investigate inherited and acquired platelet disorders, a platelet aggregation test using light transmission aggregometry (LTA) with platelet-rich plasma (PRP) was performed [51]. Platelet aggregation is detected by measuring light transmission through a sample of PRP obtained by whole blood centrifugation. The LTA method relies on the PRP property of being a turbid suspension of cells that significantly reduces light transmission. Following agonist addition, platelet aggregates reduce the turbidity of the sample, resulting in increased light transmission (100% light transmission is set with platelet-poor plasma). The dynamics of platelet aggregation (expressed as %) is therefore measured in real time as platelets aggregate. Some lumi-aggregometers can also simultaneously measure aggregation by LTA and secretion of nucleotides by luminescence [52]. LTA can provide aggregation traces induced by numerous agonists and agonist concentrations, thus offering the possibility of studying the effects of different platelet activation pathways. Also, for this reason, over the past 60 years, LTA has been the preferred platelet function test in most specialized laboratories as it correlates with bleeding and ischemic outcomes [53,54].

Although LTA is still regarded as the gold standard method and the most commonly used for the identification of many platelet disorders [55,56], it is time-consuming, poorly standardized, and requires expertise and stringent manipulation for sample processing.

### 4.2. Whole Blood Aggregometry

Impedance aggregometry allows for the study of platelet aggregation in anticoagulated whole blood, requiring a small quantity of whole blood, and no manipulation before testing, thus preserving as much as possible the physiological condition [57]. However, results from whole blood aggregometry can be useful for the diagnosis of severe platelet disorders but are insufficient to discriminate between mild platelet function disorders. A computerized whole blood aggregometry instrument (Multiplate) has gained general favor in clinical laboratories as it is fast, easy-to-use, fully automated, requires a small sample size, and uses disposable cuvettes/ electrodes with a range of diverse agonists for different applications, including monitoring of antiplatelet therapy [58,59]. Even though the basic methodologies for impedance aggregometry and Multiplate are similar, the results from the use of Chrono-Log (in Ω, as the maximal amplitude of impedance achieved) and Multiplate (in arbitrary units, as the area under the curve achieved over 6 min of aggregation) are different.

### 4.3. VerifyNow

The VerifyNow instrument (Werfen, Barcelona, Spain) is a fully automated point-of-care test specifically developed to monitor the effects of antiplatelet drugs, such as aspirin, clopidogrel, and GPIIb-IIIa antagonists, in whole blood samples [60]. It relies on the consequence of the interaction between fibrinogen and the activated GPIIb-IIIa complex, based on agonist-induced agglutination of artificial microbeads covered with fibrinogen. In cases of aspirin intake, the agonist used is arachidonic acid, whereas ADP combined with prostaglandin E1 (PGE1) is used to monitor the effect of clopidogrel, and thrombin receptor-activating peptide (TRAP) is used to monitor the antiplatelet effect of GPIIb-IIIa antagonists [60]. After agglutination following stimulation, the increase in light transmission through the sample is converted into aspirin response units (ARU) for the aspirin cartridge, P2Y12 response units (PRU) for the P2Y12 cartridge, and platelet aggregation units (PAU) for the GPIIb-IIIa cartridge. The VerifyNow instrument does not require a specialized laboratory, and it can provide immediate information. Factors influencing the performance of the assay include platelet count, hematocrit, blood triglyceride, fibrinogen levels, and its inflexibility [61].

### 4.4. Platelet Function Analyzer-100/200

The Platelet Function Analyzer (PFA)-200 (Siemens Healthcare Diagnostics Products, Munich, Germany) is an update of the first released PFA-100, relying on high-shear platelet adhesion and aggregation during the formation of a platelet plug. The instrument measures the time, reported as closure time, required to occlude an aperture in a membrane coated with collagen and ADP or collagen and epinephrine. Since the two cartridges have been shown to be largely unsensitive to P2Y12 receptor inhibitors, a third INNOVANCE P2Y cartridge, containing a membrane coated with ADP and PGE1 supplemented with calcium, has been performed to detect this class of antiplatelet drugs [62,63,64]. A platelet function test using the PFA system is simple and rapid, does not require substantial specialist training, and requires a small blood sample. However, the closure time is sensitive to platelet count, hematocrit, and vWF variation due to the high-shear conditions within the cartridge capillary and aperture. Furthermore, this kind of analysis is inflexible due to fixed stimulus concentrations.

### 4.5. Thromboelastography

Thromboelastography (TEG) relies on the whole blood clotting method. Measured values give information on the platelet contribution to clot stiffness and platelet dysfunction. Anticoagulated whole blood is incubated in a heated sample cup containing a suspended pin oscillating in each direction and connected to a computer [65]. In normal conditions, the anticoagulated blood sample does not affect the pin, whereas in the presence of blood clots, any impediment in cup motion is transmitted to the pin and recorded. TEG provides various data relating to the clot process and fibrinolysis. Rotation thromboelastometry (ROTEM) is an adaptation of TEG. The newer ROTEM platelet relies on impedance aggregometry, similar to multiple electrode aggregometry (MEA), specifically used to investigate the effect of anti-platelet drugs [66]. Since LTA is not feasible in acute situations as it is labor-intensive and requires expertise regarding its execution and interpretation, both ROTEM and ROTEM platelets provide simultaneously fast results on coagulation and platelet function. Given the significant interlaboratory variation and the poor sensitivity to various aspects of platelet function, they are not routinely recommended for platelet function testing.

### 4.6. VASP Phosphorylation

The measurement of vasodilator-stimulated phosphoprotein (VASP) phosphorylation levels by flow cytometric assay provides the in vivo effect of thienopyridines (clopidogrel, ticlopidine, and prasugrel) on platelet response. As mentioned, these drugs inhibit selectively and irreversibly ADP’s ability to activate the P2Y12 receptor. VASP is an intracellular platelet protein that is non-phosphorylated at the basal state and modulated by cAMP levels [12].

Prostacyclin activates the AC/cAMP/PKA pathway, which is, in contrast, inhibited by ADP through P2Y12 receptors. Thus, the levels of VASP phosphorylation correlate with the degree of P2Y12 receptor inhibition, whereas the VASP non-phosphorylation state correlates with the P2Y12 receptor in its active form [67,68]. The assay can be performed independently of the platelet count [69,70]. However, the cost, the requirement for a specialized technician, and the interpretation of results are somewhat subjective and make the results different from one laboratory to another.

## 5. Evidence Regarding Different Antiplatelet Agents

Even though ASA still represents the most prescribed antiplatelet drug, the panel of available agents has expanded throughout the years to overcome its limitations. We present an overview of the most commonly used antiplatelet therapies (Table 1), highlighting the pharmacological properties of each agent and presenting evidence regarding their use in the context of ACS.

### 5.1. Aspirin or Acetylsalicylic Acid (ASA)

First discovered in 1798 as a component of willow bark to treat malarial symptoms, growing technological innovation led to the production of the first aspirin tablet in 1904 [71]. The Nobel-awarding discovery of prostaglandin synthesis’ inhibition by anti-inflammatory drugs (including aspirin) and the potential to prevent thromboxane A2-mediated platelet aggregation and activation suggested a role for aspirin in the treatment of cardiovascular disease, namely, acute myocardial infarction. The survival benefit related to aspirin therapy was first demonstrated by the ISIS-2 Trial [72], which highlighted how 162.5 mg short-course aspirin therapy, administered within 24 h from symptoms’ onset in patients with suspected acute myocardial infarction (AMI), led to a significant reduction in 35-day vascular mortality, a benefit that persisted at long-term follow up.

In 1985, the Food and Drug Administration (FDA) approved aspirin use for treatment and secondary prevention of AMI, a role confirmed by the latest European Society of Cardiology (ESC) and American College of Cardiology/American Heart Association (ACC/AHA) documents for the management of ACSs, both of which include a class I recommendation for its use in the context of a standard 12-month DAPT regimen [1,2]. A load dose (LD) of 300–325 mg, followed by a maintenance dose (MD) of 75–100 mg/die, has proven to be effective in patients undergoing invasive evaluation [73,74].

Despite the historical role of aspirin and its latest recommendations, starting in the early 2010s, a growing number of randomized clinical trials (RCTs) have challenged the concept of 12-month DAPT [75,76,77,78], yielding promising results when considering specific cohorts of patients at high bleeding risk (HBR) or undergoing non-complex PCI [79,80]. The evolution of stent technology, together with the increasing frailty of patients undergoing revascularization, has shifted the focus from thrombotic to hemorrhagic risk. Stent thrombosis has a strong impact on mortality, especially in the first month following PCI [81]. During this period, it is fundamental to deliver strong antithrombotic therapy despite bleeding risk. Afterwards, ischemic burden gradually decreases while hemorrhagic risk endures, resulting in increased non-cardiovascular mortality, especially in patients with HBR [82]. This explains the search for antiplatelet therapy’s alternatives to optimize treatment safety while providing protection from ischemic events, such as DAPT shortening and early aspirin discontinuation followed by P2Y12i monotherapy [83,84,85]. RCTs testing this strategy versus standard DAPT have shown similar stent thrombosis and major adverse cardiovascular events (MACEs) incidence, with a significant reduction in major bleeding events. As a result, even though 12-month DAPT is still recommended for patients after ACS, the ESC 2020 NSTE-ACS guidelines introduced a new therapeutic option for HBR patients: DAPT discontinuation and P2Y12i monotherapy after 3 months [86]; ESC 2023 ACS guidelines state that P2Y12i monotherapy should be considered in event-free patients after 1 month DAPT for HBR patients or 3-6 month DAPT for HBR and non-HBR patients [1].

### 5.2. Clopidogrel

Clopidogrel is a second-generation thienopyridine that irreversibly inhibits the P2Y12 receptor. After demonstration of its superiority compared to aspirin in preventing ischemic events in patients with clinically manifested atherosclerotic disease, the role of clopidogrel was confirmed as part of efficacious DAPT in reducing the risk of MACEs in medically managed or proceeding to PCI ACS and after elective PCI [87,88,89,90,91]. Considering the arising evidence regarding the benefits of its use together with its limited side effects, clopidogrel became the standard thienopyridine for the DAPT regimen, replacing ticlopidine. Being a prodrug, clopidogrel requires hepatic metabolism by the cytochrome P450 family 2 subfamily C member 19 (CYP2C19) enzyme to be converted into its active form (Figure 3A).

Together with the slow onset of action, variable antiaggregant responses to clopidogrel have led to the preference of faster and more powerful P2Y12i, such as prasugrel or ticagrelor, whose use is recommended over clopidogrel in the latest guidelines [1]. However, clopidogrel still holds an important spot in several clinical settings.

In particular, clopidogrel is the first choice as P2Y12i in patients who undergo PCI and have an indication to receive long-term anticoagulation [1]. The optimal combination of anticoagulant and antiaggregant therapy is far from defined, and triple antithrombotic therapy (TAT), consisting of an oral anticoagulant combined with DAPT, drastically increases the risk of major bleeding. Starting in 2016, the PIONEER-AF, RE-DUAL PCI, AUGUSTUS, and ENTRUST-AF PCI trials showed an outperformance of direct oral anticoagulants over vitamin K antagonists (VKA) in patients with atrial fibrillation undergoing PCI, due to lower rates of major bleeding while avoiding a significant increase in ischemic events [92,93,94,95].

When comparing dual and triple antithrombotic therapy, a large metanalysis has shown a significant reduction (rate ratio [RR], 0.66; 95% confidence interval (CI), 0.56–0.78; *p* < 0.0001) in major and non-major bleeding without an increase in death, cardiovascular (CV) death, and MACE in patients receiving dual antithrombotic therapy (DAT) versus TAT; however, a higher rate of stent thrombosis (1.0% vs. 0.6%; *p* 0.04) was highlighted [93]. Definite evidence regarding the need for and the optimal duration of TAT is still lacking. Current guidelines recommend a 1-week TAT regimen with aspirin, clopidogrel, and a direct oral anticoagulant (DOAC), potentially extendable to 1 month in cases of high ischemic risk. The use of ticagrelor or prasugrel, which were poorly prescribed in the former trials’ population and were associated with increased hemorrhagic events, is discouraged [1].

### 5.3. Prasugrel

Prasugrel is a third-generation thienopyridine. Despite being a prodrug like clopidogrel, it shows a more rapid and efficient generation of active metabolites (Figure 3B), resulting in more predictable inhibition of platelet aggregation [96]. A head-to-head comparison of second and third-generation thienopyridines was performed in the TRITON-TIMI 38 trial [97], which randomized more than 13,000 patients with moderate-high-risk ACS (25% STEMI) proceeding to PCI to receive either prasugrel (60 mg loading dose and then 10 mg daily) or clopidogrel (600 mg loading dose and then 75 mg daily) on top of aspirin with a medium follow-up of 14.5 months. There was a significant reduction in the primary endpoint (cardiovascular death, non-fatal myocardial infarction, and non-fatal stroke; HR, 0.81; 95% CI, 0.73 to 0.90; *p* < 0.001), which was mainly driven by a reduction in recurrent myocardial infarction (MI). Nonetheless, the benefits in ischemic outcomes were associated with a substantial increase (32%) in thrombolysis in myocardial infarction (TIMI) major bleeding events. This increase involved both fatal and life-threatening bleeding events, as well as intracranial hemorrhages in patients with prior stroke or transient ischemic attack (TIA). In the latter group of patients, the balance between bleeding events and ischemic outcomes suggested a net clinical harm deriving from the use of prasugrel, which should not be prescribed in this cohort of patients. On the other hand, prasugrel resulted in a net clinical benefit in other ACS patients proceeding to PCI and is now recommended over clopidogrel in this setting. Other two categories of patients were associated with the absence of net clinical benefit: elderly (>75 years old) and low-weight (<60 Kg) patients. In these categories, a low maintenance dose (5 mg) is recommended [98,99].

Finally, a prasugrel-based DAPT regimen was tested in medically managed ACS patients. The TRILOGY-ACS Trial [100] included 9326 patients with high-risk NSTE-ACS who did not proceed to PCI, comparing DAPT with prasugrel (daily dose adjusted according to age and weight as suggested by the TRITON TIMI 38 Trial) versus clopidogrel. Results showed no significant difference in the primary outcome (CV death, MI, and stroke) between the two groups but showed a reduction in repeated ischemic events in the prasugrel group (HR for <12 months from ACS index event, 0.94 [95% CI, 0.79 to 1.12], vs. HR for ≥12 months, 0.64 [95% CI, 0.48 to 0.86]). At the same time, there was no increase in fatal, life-threatening, and major bleeding events; an increase in moderate and minor bleeding events was registered, accounting for more intense and efficacious platelet activity’s inhibition.

### 5.4. Ticagrelor

Differently from clopidogrel and prasugrel, ticagrelor is a cyclopentyl-triazolopyrimidine that directly and competitively inhibits the P2Y12 receptor without requiring conversion to an active metabolite (Figure 3C). Therefore, it displays both a rapid onset and offset of its antiaggregant effect, requiring bis in die administration.

The PLATO trial [101] was a landmark study directly comparing ticagrelor and clopidogrel in 18,624 patients with ACS, both of whom proceeded to PCI and were treated medically. DAPT with aspirin and ticagrelor was associated with a 16% reduction (9.8% vs. 11.7%; *p* < 0.001) in the primary outcome consisting of vascular death, MI, and stroke. Except for stroke, a significant reduction in the single components of the primary outcome was registered in the ticagrelor group. Benefits from ticagrelor therapy were confirmed in ACS patients proceeding to invasive treatment, being associated with a reduction in stent thrombosis. Considering the greater platelets’ inhibition, an increase in bleeding events was expected, as previously observed when comparing prasugrel and clopidogrel. Even though the results did not show a significant increase in the rates of major bleeding, ticagrelor was associated with an increase in non-coronary artery bypass grafting (CABG)-related major bleeding and non-fatal and fatal intracranial hemorrhages. Dyspnea was the most common side effect (13.8% of patients receiving ticagrelor), but it led to the suspension of treatment in a minority of cases. Notably, ticagrelor was associated with a 22% reduction in 1-year death from any cause, a result that had not been registered when exploring the benefits of other P2Y12i. Jointly, the trial suggested an expansion of ticagrelor’s benefits, which persisted independently of type of ACS, and resulted in a change in guidelines, which recommend ticagrelor (or prasugrel) rather than clopidogrel as P2Y12i in ACS patients. However, several studies have challenged some of the trial conclusions and questioned the real benefit of ticagrelor, particularly in high-bleeding-risk populations such as East Asians [102] or the elderly [103,104]. In the POPular AGE trial [105], involving >1000 patients aged > 70 and with NSTE-ACS, clopidogrel resulted in a reduction in bleeding events without an increase in net adverse clinical events (NACE). In a cohort study including >5000 ACS patients, comparison between clopidogrel and ticagrelor did not show any significant difference in all-cause death after logistic regression model adjustment and propensity score matching, not only in the overall population but also in HBR and high ischemic risk (HIR) individuals. The overall mortality in the ticagrelor group was doubled compared to the one registered in the PLATO trial, possibly reflecting the selection of a higher risk population [1].

Altogether, when it comes to antiplatelet strategy, a one-fits-all approach has been excluded, and tailoring treatment according to the patient’s bleeding and ischemic risk, comorbidities, economic status, and expected adherence to prescriptions is detrimental to optimizing long-term outcomes. Balancing between bleeding and ischemic risk can be challenging in clinical practice, as the overlapping of bleeding and ischemic risk factors is frequent. Furthermore, notorious risk factors for bleeding events, such as anemia and chronic kidney disease, are implicated in the increase in ischemic events. According to recently published guidelines, ticagrelor plays a role in strategies targeting both HBR and HIR patients. Since patients with coronary artery disease are at risk of late, recurrent ischemic events impacting patient outcome, several studies have explored whether there could be a net clinical benefit deriving from DAPT prolongation beyond 12 months. Trying to answer this question, the DAPT study [106] enrolled >9000 patients and tested prolongation of DAPT with either 75 mg/die clopidogrel or 10 mg/die prasugrel for 18 months after 12-month standard therapy. Prolongation of DAPT led to a significant reduction in MACE, but at the cost of increased moderate and major bleeding events. The results of the trial suggested that specific subgroups of patients, namely those with a history of MI, would have a greater benefit from DAPT prolongation. Consequently, the PEGASUS-TIMI 54 trial [107] selectively enrolled patients with a MI 1 to 3 years earlier, aged > 50, and with an additional risk factor, such as, age ≥ 65, diabetes mellitus requiring medication, a second prior MI, multivessel CAD, or chronic renal dysfunction. The trial involved >21,000 patients randomized to receive prolonged DAPT with either 90 mg bid or 60 mg bid ticagrelor on top of aspirin versus placebo. Both regimens, including prolongation of ticagrelor, showed a significant reduction (15%) in MACE at 33-month follow-up. The higher tolerability of treatment in terms of common side effects (dyspnea) and bleeding events led to the selection of the reduced-dose (60 mg) regimen for prolongation of DAPT, as it has been eventually included in clinical guidelines.

### 5.5. Cangrelor

Cangrelor is an intravenous ATP analog that acts as a reversible antagonist for the P2Y12 platelet receptor, resulting in a strong and rapidly reversible inhibition of platelets’ aggregation and activation [108] (Figure 3D). Considering its route of administration and rapid onset and offset of action, it immediately appeared suitable for periprocedural use. First trials evaluating the impact of cangrelor infusion during both elective and urgent PCI, followed by administration of 600 mg clopidogrel, were interrupted for futility as ad-interim analysis did not show superiority for the composite primary outcome of death, myocardial infarction, or ischemia-driven revascularization at 48 h compared to clopidogrel alone [109,110]. Results were not influenced by the time of administration of the clopidogrel loading dose (at the beginning of the procedure in the CHAMPION PCI Trial and at its end in the CHAMPION PLATFORM trial). However, a significant reduction in the incidence of stent thrombosis was registered in the CHAMPION PLATFORM trial. When reconsidering trials’ results after the introduction of the third universal definition of MI, a pooled analysis of the two RCTs displayed a significant reduction in the primary outcome (18%) as well as of the incidence of stent thrombosis (56%) in the cangrelor group, without a relevant increase in major and life-threating bleeding. According to these results, the CHAMPION PHOENIX double-blind, placebo-controlled trial enrolled >11,000 patients undergoing PCI across the whole spectrum of coronary syndromes to receive either a bolus of cangrelor followed by at least a 2 h infusion and a subsequent 600 mg or 300 mg loading dose of clopidogrel or clopidogrel alone before proceeding to PCI. The trial confirmed a 44% reduction in the primary endpoint (a composite of death, myocardial infarction, ischemia-driven revascularization, or stent thrombosis at 48 h), mainly driven by a reduction in MI and stent thrombosis, without significant differences in the primary safety outcome of severe bleeding at 48 h. Results at 48 h and 30 days were confirmed by a subsequent pooled analysis of the three trials [111]. Based on these results, cangrelor may be considered in P2Y12i-naïve patients proceeding to PCI, assessing case by case [1]. Furthermore, due to its pharmacological properties, it can be used in patients requiring platelet inhibition and candidates for surgery as a bridge therapy to mitigate thrombotic and hemorrhagic risk [112]. The introduction of an intravenous antiaggregant has further expanded the possible scenarios when it comes to the choice of antiplatelet therapy. On top of identifying which option is the most suitable for the patient, it is not uncommon to have to switch from one P2Y12i to another for clinical reasons. Initial concerns regarding the safety of switching, particularly in the setting of the early phase after PCI in ACS, have prompted an effort to produce clear consensus documents [113] to guide physicians. The pharmacological features of each agent, together with the time interval from the index event, are to be considered when establishing when and whether to administer a loading dose of the new antiplatelet agent. Following recommendations, switching is feasible while ensuring safety and efficacy.

### 5.6. Comparing Prasugrel and Ticagrelor

According to the latest guidelines, 12-month DAPT with aspirin and a potent P2Y12i is the standard therapy for patients with ACS not at HBR, and a series of studies have been trying to identify whether one is superior to the other. The PRAGUE-18 trial [83,84] comparing Prasugrel versus Ticagrelor in >1000 patients with STEMI undergoing primary PCI was stopped prematurely for futility, as it did not show any significant difference in the primary outcome consisting of death, re-infarction, target vessel revascularization, bleeding requiring transfusion, and prolonged (>7 days) hospital stay, as well as in the incidence of bleeding academic research consortium (BARC) and TIMI major bleeding events. In the study, a significant proportion of patients (34.1% and 44.4% in the prasugrel and ticagrelor groups, respectively) switched to clopidogrel due to economic issues. Interestingly, as this was done after consultation and agreement with the treating physician and allowed only in low-risk patients, it did not result in an increase in ischemic events. However, switching to clopidogrel could have limited the head-to-head comparison of the two drugs.

Originally designed to assess superiority of ticagrelor over prasugrel in patients presenting with ACS proceeding to invasive evaluation, the ISAR REACT 5 Trial [114] led to unexpected results, as it showed an excess of the primary end-point of death, myocardial infarction, and stroke at 12 months (HR 1.36; 95% CI, 1.09 to 1.70; *p* = 0.006) in the ticagrelor group compared to prasugrel, mainly led by an excess of myocardial infarctions (4.8% in the ticagrelor group and 3.0% in the prasugrel group; HR 1.63, IC 95% 1.18–2.25) and without any difference in BARC 3–5 bleeding events. Based on the results of this trial, ESC guidelines suggest that prasugrel should be preferred to ticagrelor in ACS patients proceeding to PCI [1]. However, when interpreting the unexpected results of the trial, some critical issues should be considered. The results in the primary endpoint were mainly led by a reduction in MI in the prasugrel group. However, if the incidence of MI was similar in patients receiving ticagrelor in the PLATO and ISAR REACT 5 trials, a substantial difference was observed between the TRITON-TIMI 38 and ISAR REACT 5 populations, as the incidence of MI was more than halved in patients receiving prasugrel in the latter, a phenomenon that can be partly explained by the different definitions of MI (the investigators referred to the 3rd universal definition of MI [115]). Furthermore, there appears to be an underestimation of non-adherence to treatment (0.9% for the prasugrel group and 0.4% for the ticagrelor group), as there was no monitoring of patients’ compliance with antiplatelet treatment. Both groups displayed a significant proportion of drug discontinuation (15.2% in the ticagrelor group and 12.5% in the prasugrel group), with a median interval from discharge to discontinuation of 84 days for ticagrelor and 109 days for prasugrel (*p* = 0.01), a difference that could contribute to prasugrel’s favorable results and be explained by ticagrelor’s more common side effects, such as dyspnea.

In a network metanalysis, including more than 56,000 patients and 12 RCTs [116], Navarese et al. did not find any significant difference when directly comparing prasugrel and ticagrelor. However, prasugrel proved to reduce overall MI when compared to clopidogrel, while ticagrelor led to a reduction in spontaneous MI rather than periprocedural MI and failed to dictate any difference in its overall incidence. On the other hand, only ticagrelor did show a reduction in overall mortality and CV death.

The on-going SWITCH SWEDEHEART trial (NCT05183178) [117], a stepped-wedged randomized trial, will further investigate the efficacy and safety of potent P2Y12i in hospitalized ACS patients, hopefully shedding light on the advantages, if any, of one drug compared to the other.

Finally, it should be remembered that the trial compared different pharmacological strategies: pre-treatment with ticagrelor (as the load dose was administered at the time of diagnosis) and no pre-treatment with prasugrel (the load dose was administered at the time of PCI). According to the results of the ATLANTIC Trial [118], ESC 2023 ACS guidelines downgrade pre-treatment with P2Y12i in the STEMI population (Class of recommendation IIb), while routine pre-treatment with P2Y12i is not recommended [119].

## 6. Alternative Strategies for Patients with High Bleeding Risk

### 6.1. P2Y12i Monotherapy

Considering the relevant impact of bleeding events in patients receiving DAPT, there has been an increasing effort to identify alternative antiplatelet strategies with the potential to guarantee protection from ischemic events while reducing bleeding complications (Figure 4).

As DAPT is inevitably associated with an increase in bleeding risk, it was first tested to see whether patients could benefit from a shorter DAPT regimen without an increase in ischemic events. The first trials assessing this strategy compared 1- to 6-month DAPT followed by P2Y12i early suspension and aspirin monotherapy versus 12-month standard DAPT. The MASTER-DAPT trial [79], which was the first one to focus on HBR patients, highlighted how 1-month DAPT after PCI and implantation of drug-eluting stents (DES) was superior to 3-month DAPT in reducing major bleeding events and non-inferior in NACE and major adverse cardiovascular and cerebrovascular events (MACCE). These results were consistent independently of clinical presentation (ACS or chronic coronary syndrome (CCS)) and complexity of coronary lesions.

Given the promising results of early aspirin discontinuation in patients with ACS requiring antiplatelet therapy as well as oral anticoagulation, a growing number of trials have started testing the hypothesis that DAPT shortening with early aspirin’s discontinuation followed by P2Y12i monotherapy could lower bleeding risk (particularly gastrointestinal bleeding related to aspirin’s well-known toxicity) while ensuring protection from ischemic events. The GLOBAL LEADERS trial [84] was the first to test this alternative strategy. It included >15,000 patients and compared 1-month DAPT followed by 23-month ticagrelor monotherapy versus 12-month DAPT followed by aspirin monotherapy. The P2Y12i monotherapy strategy was not superior in preventing the primary outcome of all-cause death and non-fatal MI and showed a similar frequency of major bleeding. Subsequently, the TWILIGHT and the TICO trials evaluated ticagrelor monotherapy after 3-month DAPT in different settings. The TWILIGHT Trial included >7000 patients proceeding to primary or elective high-risk PCI. In the absence of ischemic events, after 3-month DAPT with ticagrelor on top of aspirin, patients were randomized to receive ticagrelor plus aspirin versus placebo. The study included only HIR patients, based on the presence of at least one clinical and one PCI ischemic risk feature, and showed a significant reduction (44%) in the primary outcome of BARC 3–5 bleeding in the ticagrelor monotherapy group without an increase in MACE. Results were confirmed across the whole spectrum of clinical presentation. The open-label TICO trial [85] enrolled >3000 ACS patients proceeding to PCI after 3-month ticagrelor-based DAPT and was randomized to either receiving ticagrelor monotherapy or continuing standard DAPT up to 12 months. The primary outcome (composite of major bleeding and NACE) was significantly reduced in patients receiving ticagrelor monotherapy. However, event rates were lower than expected. A metanalysis of trials testing P2Y12i monotherapy [1] confirmed that the benefits of short DAPT in terms of a reduction in bleeding events did not come at the cost of increased ischemic events.

However, it should be kept in mind that trials evaluating P2Y12i monotherapy with clopidogrel mainly enrolled patients with low-ischemic risk. In the STOP-DAPT 2 trial [120], >3000 Japanese patients undergoing PCI with DES implantation (<40% ACS) received either 1-month DAPT with clopidogrel followed by clopidogrel monotherapy or 12-month clopidogrel-based DAPT. The P2Y12i monotherapy strategy proved both non-inferior and superior for the composite primary outcome of cardiovascular and bleeding events. However, a few limitations prompt caution. First, the study was underpowered to assess short-term DAPT-related stent thrombosis, and statistical power was limited by low event rates. Second, the patients enrolled represented a low-ischemic risk population due to their ethnic, clinical, and coronary features. In the STOP-DAPT 2 ACS trial enrolling >4000 ACS patients, 1- to 2-month clopidogrel-based DAPT followed by clopidogrel monotherapy failed to show non-inferiority for a composite primary outcome of bleeding and cardiovascular events, with a rise in ischemic events [121].

Considering the accumulating evidence, early aspirin suspension followed by P2Y12i monotherapy does not represent the standard antiplatelet regimen in ACS patients. However, its role is recognized in HBR patients, for whom it may be considered, as well as in P2Y12i suspension followed by aspirin monotherapy after one month of DAPT [1].

### 6.2. PFT-Guided Anti-P2Y12 Therapy

RCTs aimed at verifying the role of anti-platelet therapy tailored based on the results of the platelet function test (PFT) in patients undergoing PCI starting with standard treatment have given results that are sometimes contrasting because for PFT-guided therapy, different approaches in terms of methods, labor intensity, and endpoints were used. Most clinical trials show the effects of escalation to potentially more effective antiplatelet agents in clopidogrel patients who, at enrolment, presented LoF mutations or high on-treatment platelet reactivity (HTPR). The switch to a clopidogrel-based DAPT regimen is a valid option to be considered when bleeding events or side effects (such as ticagrelor-related dyspnea) limit the prosecution of class I recommended DAPT, including prasugrel or ticagrelor. Despite registries having already pointed out that de-escalation is quite common in clinical practice (up to 28% of patients switch from potent P2Y12i to clopidogrel within the first year after the ACS index event) [122,123], RCTs exploring de-escalation strategies and assessing their safety have emerged in the latest years, including a variable proportion of ACS patients and testing different modalities to conduct de-escalation: guided or unguided. Indeed, guided de-escalation can be achieved through PFT or genotype testing.

It is possible to identify patients who, displaying inadequate platelets’ inhibition after de-escalation, are at higher risk of ischemic events, thus requiring the prosecution of standard DAPT therapy. The TROPICAL-ACS trial [124] included more than 2500 patients with biomarker-positive ACS proceeding to successful PCI and aimed to receive 12-month DAPT with prasugrel as P2Y12i. Patients randomized in the de-escalation group received 1-week DAPT with prasugrel and aspirin, subsequently switching P2Y12i to clopidogrel. At a 2-week follow-up visit, a PFT was carried out, and only patients with sufficient platelet inhibition prosecuted de-escalation; patients with HPR switched back to prasugrel. The trial demonstrated the non-inferiority of DAPT de-escalation for net adverse clinical events (NACE) without any difference in bleeding outcomes. However, it should be highlighted that nearly 40% of patients in the de-escalation group showed HPR at 1-week PFT, requiring a switch back to prasugrel.

Indeed, a number of RCTs, including patients showing HTPR phenotype based on platelet function screening and randomized to a different antiplatelet regimen, have been performed [125,126,127,128,129,130,131,132,133,134,135,136,137,138,139,140,141,142,143,144].

Individually, in most cases, no significant benefit was demonstrated in terms of reduction in adverse cardiac events or bleeding when the PFT-guided approach was compared to standard care. Among them, GRAVITAS [125] and TRIGGER-PCI studies [129] were focused on HTPR patients randomized to different antiplatelet therapies, whereas the others included patients with or without HTPR then randomized to PFT-guided or standard therapy. Notably, as already mentioned, TROPICAL-ACS met the non-inferiority hypothesis on net cardiac outcomes with no benefit on bleeding, and PATH-PCI met the superiority hypothesis on net adverse cardiac events with no advantage on bleeding [124,145]. However, the findings of these two clinical trials were criticized for some methodological approaches [146]. Multiple meta-analyses were performed to overcome the relatively low power of each study and to determine whether PFT-guided therapy can actually be useful for detecting benefit or harm at hard clinical end points. In general, these pooled analyses indicate that PFT-guided therapy, generally performed by using PFT such as VerifyNow, LTA, VASP phosphorylation, TEG, or multiplate assays, in comparison with non-guided therapy, is a tool contributing to reducing the risk of MACE with little or no influence on bleeding risk [147,148,149,150,151,152,153,154]. However, a recent systematic review has performed three distinct meta-analyses, each exploring only RCTs with a homogeneous study design and considering the geographical regions in which RCTs were performed [155]. The overall analysis of RCTs revealed that PFT-guided therapy provides no significant benefit to the incidence of MACE or bleeding events in comparison with the standard treatment group. However, the sub-analysis of pooled results from RCTs performed only in China revealed that the risk of MACE was significantly lower in the PFT-guided arm, despite the low statistical power of these studies [155]. According to this study, the impact of RCTs aimed at verifying the relative risk for clinical endpoints of PFT-based anti-platelet therapy is deeply influenced by criteria applied to compare different study designs and geographical areas in which the clinical study is carried out. Another recent meta-analysis, including 7691 patients from five RCTs, has confirmed that using PFT to adjust antiplatelet treatment does not improve clinical outcomes, and a lack of clarity on PFT utility may explain why present guidelines do not currently recommend its routine use in the ACS context [156].

Collectively, although PFT has promising results, its implementation in everyday clinical practice is challenging. More user-friendly tests are now available, but they are not free from limitations regarding the variability of results. In conclusion, although many of these tests have potential clinical utility, large RCTs are still needed to determine whether routine platelet testing may be useful to adapt or titrate antiplatelet treatment based on the results of a platelet function assay. In our opinion, implementation of these tests could benefit specific categories of patients with higher hemorrhagic or thrombotic risk, as in the case of multi-vessel coronary disease requiring complex PCI with implantation of multiple stents or in elderly patients with fragility conditioning high bleeding risk.

### 6.3. Genotype-Guided Anti-P2Y12 Therapy

Genetic variability regarding the CYP2C19 isoform has been proposed as a determinant of variable platelet reactivity under clopidogrel therapy. The most important polymorphism affecting clopidogrel metabolism is the CYP2C19*2 LoF polymorphism, which results in the virtual absence of clopidogrel’s active metabolite in homozygotes. CYP2C19*3 LoF polymorphism is another common cause of reduced clopidogrel activation in the Asiatic population, where it can be found in up to 10% of patients; despite the diffusion of genetic variants resulting in HPR, East Asians do not show an increased ischemic risk and are, in contrast, more sensitive to bleeding complications, a phenomenon that has been described as the “Asian paradox” [157]. This highlights how genetic background should not be interpreted as a surrogate for PFT, as it is a single element contributing to the complex, multi-factorial response to clopidogrel. However, when assessed through validated assays, CYP2C19 genotyping can help identify patients who, being carriers of LoF alleles, would be at risk of HPR-related thrombotic complications during de-escalation of antiplatelet therapy. Genetic testing was used to select STEMI patients for de-escalation of DAPT after primary PCI in the POPular Genetics Trial [158]. Genotype-guided de-escalation strategy proved to be non-inferior for NACE and superior for major and minor bleeding compared to standard DAPT (9.8% vs. 12.5%; hazard ratio, 0.78; 95% CI, 0.61 to 0.98; *p* = 0.04). The difference in the primary bleeding outcome (PLATO major and minor bleeding events) was driven by a reduction in minor bleeding in the de-escalation group. Nonetheless, the reduction in minor bleeding events is clinically and economically relevant, as they contribute to the suspension of treatment and health costs. Genetic phenotyping is far from decoded; apart from well-known LoF polymorphisms, rarer genetic variants could play a role in defining thrombotic as well as bleeding risk, but evidence regarding their impact is conflicting. The fact that genotyping mainly focuses on well-described LoF polymorphism, together with evidence of low sensitivity for predicting high-on treatment platelet reactivity when compared to pharmacodynamic assays, has raised uncertainty about the urgency of integrating its use in clinical practice [159].

### 6.4. Unguided De-Escalation of P2Y12i

Several trials have tested unguided de-escalation in patients with ACS after a 1-month standard DAPT. In the TOPIC trial [160], unguided de-escalation proved to be associated with a significant reduction in NACE (which did not include myocardial infarction) and bleeding events (BARC 2–5). The TALOS-AMI trial [161] confirmed superiority in the NACE outcome of unguided de-escalation to clopidogrel after 1-month DAPT with aspiring and ticagrelor in more than 2500 East Asian patients with ACS. Switching to clopidogrel was associated with a reduction in bleeding without increasing ischemic events. Finally, the HOST-REDUCE-POLYTECH-ACS trial [162] tested another strategy for de-escalation: an unguided switch from the standard dose (10 mg daily) to the reduced dose (5 mg daily) of prasugrel after 1-month DAPT with standard dose prasugrel in ACS East Asian patients following PCI. The trial showed a significant reduction in the primary outcomes of NACE and BARC 2–5 in the de-escalation arm. Caution is needed when extending results obtained in the East Asian population to other ethnicities, as they are more prone to bleeding than ischemic complications.

Overall, the latest guidelines do recognize a role for de-escalation of antiplatelet therapy from prasugrel/ticagrelor to clopidogrel, which may be considered an alternative strategy to reduce bleeding risk after 1-month standard DAPT [1]. In the first 30 days after the ACS index event, de-escalation is contraindicated. According to the 2019 ACC/AHA PFT and genetic testing Consensus Statement, pharmacodynamic assays and genetic phenotyping should be intended as supplementary tools when personalizing the P2Y12i strategy [157].

## 7. Conclusions

Antiplatelet therapy is a cornerstone in the management of acute coronary syndromes. Even if recent guidelines confirm the use of a dual antiplatelet regimen consisting of aspirin and a P2Y12 inhibitor as the standard of care, dramatic changes in stent technology and features of patients have led to greater attention to the balance between hemorrhagic and thrombotic risk. Taking into consideration all these aspects, recent years have been characterized by an effort to define alternative strategies to limit hemorrhagic complications while ensuring an efficacious antithrombotic effect, which have been tested in numerous RCTs, such as P2Y12 monotherapy and guided or unguided downgrading of DAPT. Evidence regarding the net clinical benefit of these alternative strategies in patients with high bleeding risk has led to an update of the latest guidelines. While PFT diffusion and viability are still limited, these assays have the potential to identify patients who can benefit from a reduction in bleeding events switching to clopidogrel while preserving an adequate platelet’s inhibition. On the other hand, genetic testing aimed at identifying variants predicting high-on-treatment platelets’ reactivity is still limited to the detection of common LoF polymorphisms, questioning the urge to implement their use in clinical practice. Overall, it is evident that a one-fits-all approach to the management of antiplatelet therapy is impracticable. The choice of antiplatelet agents and duration of DAPT must be tailored according to the patient’s individual features, while assays evaluating the individual response to pharmacological agents can help balance the hemorrhagic and thrombotic risks.

## Figures and Tables

**Figure 1 ijms-25-03981-f001:**
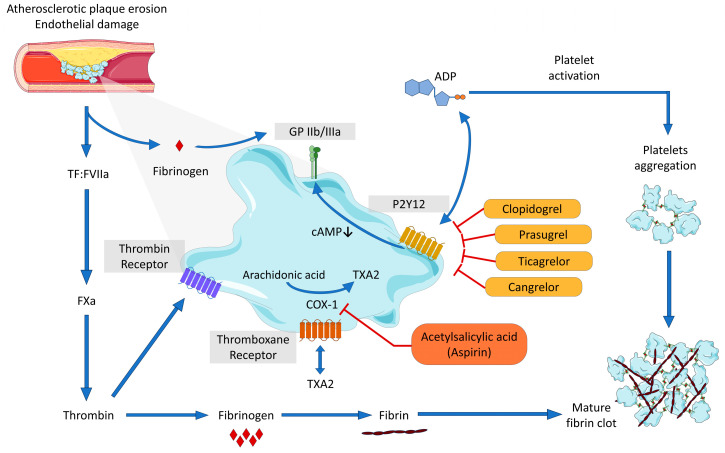
Receptors implicated in platelets’ activation and the sites of action of principal antiplatelet agents.

**Figure 2 ijms-25-03981-f002:**
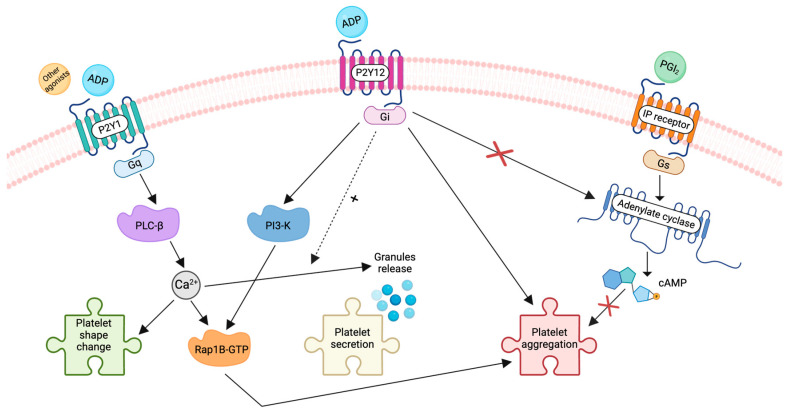
P2Y12 and P2Y1 receptors and their downstream intracellular pathways.

**Figure 3 ijms-25-03981-f003:**
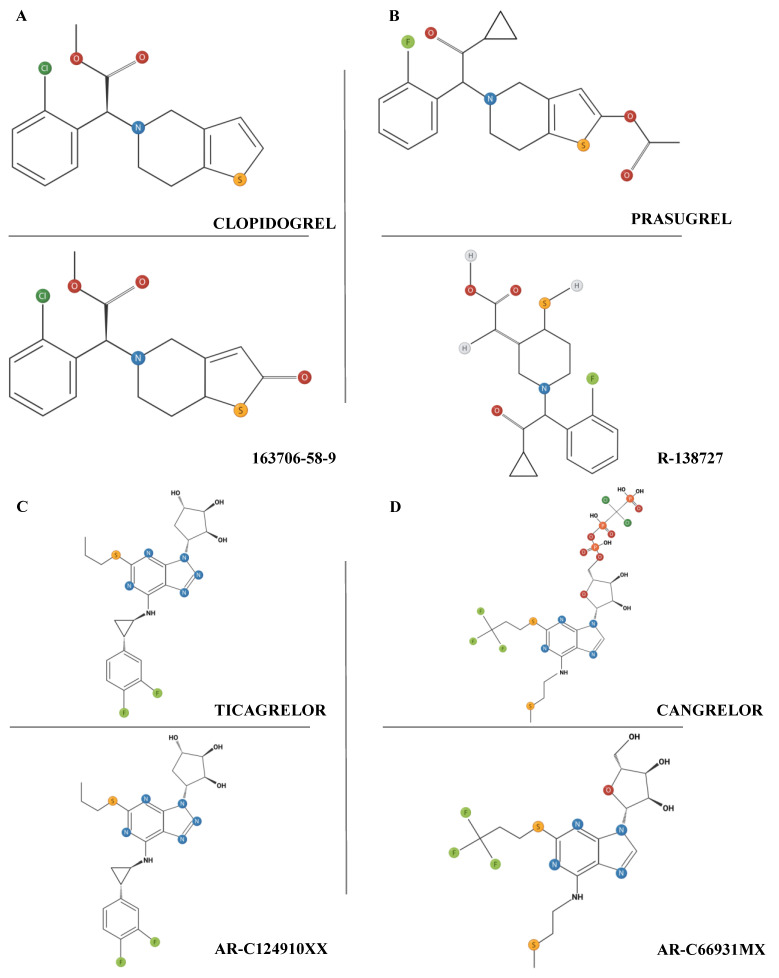
Molecular structure of the most common inhibitors of P2Y12 receptors as drugs (top of **A**–**D**) and their relative active metabolites (bottom of **A**–**D**).

**Figure 4 ijms-25-03981-f004:**
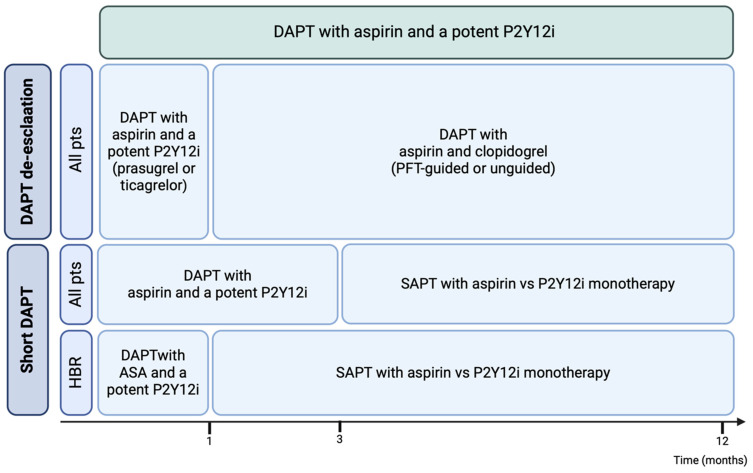
Alternative DAPT regimen to reduce bleeding risk in patients with acute coronary syndromes according to the European Society of Cardiology’s latest guidelines.

**Table 1 ijms-25-03981-t001:** Overview of the most common antiplatelet agents.

	Acetylsalicylic Acid (ASA)	Clopidogrel	Prasugrel	Ticagrelor	Cangrelor
Mechanism of Action	Inhibition of prostaglandin synthesis and TxA2-mediated platelet aggregation	Irreversible inhibition of P2Y12 receptor	Irreversible inhibition of P2Y12 receptor	Reversible inhibition of P2Y12 receptor	Reversible inhibition of P2Y12 receptor
Roat of Administration	Oral	Oral	Oral	Oral	Intravenous
Recommended Dosage in ACS	300–325 mg (loading dose) 75–100 mg Once daily	300–600 mg (loading dose) 75 mg Once daily	60 mg (loading dose) 10 mg (5 mg) Once daily	180 mg (loading dose) 90 mg Twice daily	Bolus (30 µg/Kg) followed by Infusion (4 µg/Kg/min) for at least 2 h
Half-life	20 min	≈6 h	<5 min	6–12 h	3–6 min
CYP interaction	No	CYP2C19	No	CYP3A	No
Side Effects	- Dyspepsia - Gastrointestinal bleeding	- ↑Bleeding risk - Dyspepsia - Diarrhea	- ↑Bleeding risk - Blurred vision - Dizziness	- ↑Bleeding risk - Dyspnea	- ↑Bleeding risk - Dyspnea

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
