# Peer review of "Current Strategies to Guide the Antiplatelet Therapy in Acute Coronary Syndromes"

_ijms, 2024, doi:10.3390/ijms25073981_

Round 1

Reviewer 1 Report

Comments and Suggestions for Authors

Dear Authors,

the review “New Strategies to Guide the Antiplatelet Therapy in Acute Coronary Syndromes” includes various sections.

 In this regard, some constructive considerations:

 1.      Section “description of platelets biology and the role of the main platelet signal pathways involved in platelet aggregation during an acute coronary syndrome” I think it is exhaustive and well described.

 2.      Section “the latest evidence on the evaluation of platelet function”.

The methods are well described.

4.3. VerifyNow” it is a trade name (WERFEN) that should also be specified with the name of the manufacturer or alternatively, preferably, to be identified with the method used e.g. LTA, Light-Transmission Aggregometry. The same considerations for “4.4. Platelet Function Analyzer-100/200” by Siemens Healthcare Diagnostics.

“6.2. PFT-Guided Anti-P2Y12 Therapy”: the recent meta-analysis “Role of platelet function testing in acute coronary syndromes: a meta- analysis” Open Heart 2022;9:e002129. doi:10.1136/openhrt-2022-002129 that concludes that “Compared with standard DAPT with P2Y12 antagonists, using PFT to adjust antiplatelet therapy does not improve clinical outcomes. Therefore, the positions of key guidelines on routine testing in ACS should remain unchanged. In addition, the study highlights the need for well- designed and powered RCTs and standardised testing methodologies to provide reliable findings and definitive conclusions” is not taken into account by your review.

 3.      Section “the description of the role of aspirin and P2Y12 inhibitors in the treatment of acute coronary syndromes critically appraising the available evidence from clinical trials and providing current international guidelines recommendations” I would point out that, compared to what is reported in the recent ESC 2023 guidelines on ACS, there are no news worth mentioning (that should be reported).

 Also for “the alternative therapeutic regimens to standard dual antiplatelet therapy in particular in patients at high bleeding risk” there are no major changes.

Infact, the papers not yet published at the time of printing of the ESC 2023 guidelines and reported in your bibliography are only the following, which cannot in any way modify the ESC statements, nor suggest “new Strategies to Guide the Antiplatelet Therapy in Acute Coronary Syndromes”:

50.  Chen, J.; Qu, Y.; Jiang, M.; Li, H.; Cui, C.; Liu, D. Population Pharmacokinetic/Pharmacodynamic Models for P2Y12 Inhibitors: A Systematic Review and Clinical Appraisal Using Exposure Simulation. Clin Pharmacokinet 2024, doi:10.1007/s40262-023-01335-2.

 140.                     Komócsi, A.; Merkely, B.; Hadamitzky, M.; Massberg, S.; Rizas, K.D.; Hein-Rothweiler, R.; Gross, L.; Trenk, D.; Sibbing, D.; Aradi, D. Impact of Body Mass on P2Y12-Inhibitor de-Escalation in Acute Coronary Syndromes—a Substudy of the TROPICAL-ACS Trial. European Heart Journal - Cardiovascular Pharmacotherapy 2023, 9, 608–616, doi:10.1093/ehjcvp/pvad027.

 145.                     Ammirabile, N.; Landolina, D.; Capodanno, D. Navigating the Course of Dual Antiplatelet Therapy After Percutaneous Coronary Intervention: A Review of Guided Approaches. Circ Cardiovasc Interv 2023, e013450, doi:10.1161/CIRCINTERVENTIONS.123.013450.

154.                     Birocchi, S.; Rocchetti, M.; Minardi, A.; Podda, G.M.; Squizzato, A.; Cattaneo, M. Guided Anti-P2Y12 Therapy in Patients Undergoing PCI: Three Systematic Reviews with Meta-Analyses of Randomized Controlled Trials with Homogeneous Design. Thromb Haemost 2023, a-2149-4344, doi:10.1055/a-2149-4344.

I would like to point out the part dedicated to pre-treatment, the only real novelty of recent years, in addition to the paragraph “Finally, it should be remembered that the trial compared to different pharmacological strategies: pre-treatment with ticagrelor (as the load dose was administered at the time of diagnosis) and no pre-treatment with prasugrel (load dose administered at the time of PCI). According to the results of the ATLANTIC Trial [117], ESC 2023 ACS guidelines downgrade pre-treatment with P2Y12i in the STEMI population (Class of recommenda-tion IIb) while routine pre-treatment with P2Y12i is not recommend [118].”

Other considerations:

Title: “Review - New Strategies to Guide the Antiplatelet Therapy in Acute Coronary Syndromes”.I believe, as mentioned above, that the term "new strategies" is misleading because, compared to what is described in the ESC 2023 guidelines, it is not possible to identify any new strategy that guides antiplatelet therapy in ACS.

“Introduction -  …the term identifies the simultaneous use of ac-etylsalicylic acid (ASA) and a P2Y12 inhibitor (P2Y12i), a cornerstone in the treatment of atherosclerotic cardiovascular disease in general and, more specifically, in the manage-ment of acute coronary syndromes (ACS)…(the statement is also reported in the Conclusions): aspirin and P2Y12 cannot be considered the cornerstone in the treatment of atheroslerotic cardiovascular disease (other than ACS or PCI) as the efficacy in asymptomatic atherosclerosis is not definitively elucidated.

“5.1. Acetylsalicylic Acid (ASA)” The term Aspirin is universally used instead of Acetylsalicylic Acid (ASA) but in the paper it is used alternately either one or the other: it would be useful to always use the same.

The paper is quite long (26 pages), a reduction of the same could facilitate the reading. In addition, for the same reason, it would be necessary to add some schematic table that can facilitate the understanding of the messages that you intend to send (e.g. for PFTs table with methodology used, possible clinical utility, comment with limitations).

In the conclusions section, it is impossible not to take into account, in accordance with the ESC guidelines, that “a strategy based on platelet function testing or genetic testing should be prospectively tested in patients who may benefit from de-escalating antithrombotic therapy”. In this regard, it should also be added, because it is not clear, which type of patient, in the opinion of the authors, could benefit from the use of PFT.

Reviewer 2 Report

Comments and Suggestions for Authors

The authors describe anti-platelet agents and the pathophysiology in a detailed manner with a latest review of literature

Recommend adding a table with different antiplatelets and their mechanisms along with side effect profile

Recommend also describing in detail about switching between different antiplatelets as this is something quite common in clinical practice

Comments on the Quality of English Language

Quality of language is good - no major editing needed

Reviewer 3 Report

Comments and Suggestions for Authors

The proposed manuscript is an extremely interesting one, with multiple clinical implications presented through a comprehensive review of the pathophysiological aspects associated with the topic.

The chosen topic is a topical one, with acute coronary syndromes occurring in an increasing number of patients at younger and younger ages.

The schemes developed help to understand the mechanisms of pathophysiology.

The manuscript is rigorously organized and analyses in a comprehensive manner the main aspects of the subject.

I suggest to the authors the introduction of an additional section on potential future research directions.

Reviewer 4 Report

Comments and Suggestions for Authors

The manuscript by Russo and coworkers reviewed new strategies to guide antiplatelet therapy in ACS patients. It was well written, both educational and informative, navigating rather complicated strategies to balance thrombosis and bleeding risk. It should be of interest to both researchers and clinicians and acceptable for publication after minor revisions.

1. Typo on Pg1 Ln 8 from bottom: have shifted; also on Pg 8, ln2

2. Pg8, right above Figure 3: 2C9 should be 2C19.

3. It is difficult to read the graph. Please increase the line thickness of chemical strutures.

4. Pg5, Ln5: P2Y121 should be P2Y12.
